# Predicting Interest in Orthodontic Aligners: A Google Trends Data Analysis

**DOI:** 10.3390/ijerph19053105

**Published:** 2022-03-06

**Authors:** Magdalena Sycińska-Dziarnowska, Liliana Szyszka-Sommerfeld, Krzysztof Woźniak, Steven J. Lindauer, Gianrico Spagnuolo

**Affiliations:** 1Department of Orthodontics, Pomeranian Medical University in Szczecin, Al. Powst. Wlkp. 72, 70111 Szczecin, Poland; magdadziarnowska@gmail.com (M.S.-D.); liliana.szyszka@gmail.com (L.S.-S.); krzysztof.wozniak@pum.edu.pl (K.W.); 2Department of Orthodontics, School of Dentistry, Virginia Commonwealth University, Richmond, VA 23298, USA; sjlindau@vcu.edu; 3Department of Neurosciences, Reproductive and Odontostomatological Sciences, University of Naples “Federico II”, 80131 Naples, Italy

**Keywords:** Invisalign, orthodontic treatment, esthetic appearance, dental practice management, online survey, certified

## Abstract

Aligners are an example of how advances in dentistry can develop from innovative combinations of 3D technologies in imaging, planning and printing to provide new treatment modalities. With increasing demand for esthetic orthodontic treatment, aligners have grown in popularity because they are esthetically more pleasing and less obstructive to oral hygiene and other oral functions compared to fixed orthodontic appliances. To observe and estimate aligner treatment interest among Google Search users, Google Trends data were obtained and analyzed for the search term, “Invisalign”. A prediction of interest for the year 2022 for three European Union countries with the highest GDP was developed. “Invisalign” was chosen to represent all orthodontic aligners as the most searched term in Google Trends for aligners. This is the first study to predict interest in the query “Invisalign” in a Google search engine. The Prophet algorithm, which depends on advanced statistical analysis methods, positions itself as an automatic prediction procedure and was used to predict Google Trends data. Seasonality modeling was based on the standard Fourier series to provide a flexible model of periodic effects. The results predict an increase in “Invisalign” in Google Trends queries in the coming year, increasing by around 6%, 9% and 13% by the end of 2022 compared to 2021 for France, Italy and Germany, respectively. Forecasting allows practitioners to plan for growing demand for particular treatments, consider taking continuing education, specifically, aligner certification courses, or introduce modern scanning technology into offices. The oral health community can use similar prediction tools and methods to remain alert to future changes in patient demand to improve the responses of professional organizations as a whole, work more effectively with governments if needed, and provide better coordination of care for patients.

## 1. Introduction

Technological advancements have resulted in tremendous changes in dentistry, especially orthodontics, in recent years. With growing awareness of the esthetic appearance of the face and teeth, patients are seeking orthodontic treatment to improve their appearance. Modern and more esthetically pleasing options for orthodontic treatment, innovative fully digital workflows, and new imaging methods provide patients and professionals with a new focus in orthodontic care [1]. Treatment can be accomplished using many types of appliances, including visibly inconspicuous esthetic brackets, lingual appliances, or clear aligners. Digital technology and 3D planning can help achieve natural and esthetic smiles. Artificial intelligence and machine learning developments are advancing rapidly in automated diagnostic procedures, smile design applications, and treatment planning processes [2]. New techniques and technologies in intraoral scanning, 3D printing, computer-aided design/computer-aided manufacturing (CAD/CAM) and more esthetic dental materials are likely to lead to increased demand for aligners in orthodontics [3].

According to Proffit, the ideal orthodontic appliance should not impede occlusion, obstruct hygiene, or damage oral tissues. Controlled force should be applied between visits, and good anchorage should be possible to provide desired tooth movements [4]. Currently, one of the best-known brands of clear aligners is Invisalign, introduced in 1998, and manufactured by Align Technology, Inc. (Phoenix, AZ, USA) [5]. Invisalign straightens teeth using a series of aligners made for an individual patient using digital 3D technology. The system has been used to treat 11 million patients [5]. Invisalign clear aligners are replaced by the patient every one or two weeks to move teeth gradually according to the doctor’s treatment recommendation. Correction of minor or even complex malocclusions can be accomplished to the planned and desired occlusion.

Google Trends (GT), a novel tool with anonymous and fast tracking capabilities, has been used in various scientific fields to predict public interest and can also be useful for studying and tracking medical and social behavior [6,7]. Online analysis may be an alternative to conducting large surveys with the advantage of having a shorter time lag. There have been no previous studies based on GT search query regarding predictions of future interest in orthodontic aligners. Earlier analyses of patient interest and experience with Invisalign were conducted on Twitter, in which patients expressed their opinions in real time [8,9]. A previously conducted European survey reported that the Internet was used for healthcare purposes by 71% of Internet users [10].

Given the paucity of research on the prediction of orthodontic aligner use in Europe, the aim of the study was to forecast interest in orthodontic aligners using the example of Invisalign. The purpose is to demonstrate how this methodology may be useful for predicting trends in patient demand for specific dental treatments to help practitioners and professional associations plan and allocate resources more efficiently.

## 2. Materials and Methods

The data were collected from the GT open database among anonymous Google search engine users related to orthodontic aligners [11]. The search term “Invisalign” was chosen to represent all orthodontic aligners because prevalence/frequency of the search term “Invisalign” was highest among searches on GT using the terms “aligners”, “clear aligners”, and the brands “OrthoFX”, “ClearCorrect”, “SmileDirectClub” [12]. Previous studies also selected Invisalign as the most searched term for orthodontic clear aligners and used a similar search term selection method [13]. For statistical analysis, the weekly number of queries in GT related to “Invisalign” over the last five years, starting on 8 January 2017, was collected. GT data were gathered for all categories from the web search index. Country selection was based on global gross domestic product (GDP) [14]. The number of search queries for term “Invisalign” was obtained for the European Union countries with the most developed economies by GDP: Germany, France and Italy. Each sample was normalized, ranging from 0 to 100, where 100 represented the maximum number of requests and 0 represented the low volume of queries during the specified period.

All statistical computations were made applying the “R” programming language v.4.1.1, IDE “R studio” version 1.4.1717—open-source software for data science, scientific research, and technical communication [15]. The list of packages used included the package “prophet” [16] for making predictions, and “stats” [15] for statistical testing.

For predicting GT data, the Prophet algorithm was used, which positions itself as an automatic forecasting procedure and represents a local Bayesian structural time series model. Prophet algorithm was an appropriate tool for analyzing the GT data because it is based on an additive model that fits nonlinear trends with annual, weekly seasonality, and holiday effects. This was especially important for predicting non-stationary data [17] which, in the vast majority of cases, characterized the data from GT. It works best with multiple seasons of historical data and generally handles outliers well [18].

The principle of the Prophet algorithm is based on the decomposition of the time series data by a three-component model (trend, seasonality, and holidays) according to the estimation procedures for structural time series models [19].
(1)$$y\left(t\right)=g\left(t\right)+s\left(t\right)+h\left(t\right)+\epsilon_{t},$$
where *g*(*t*) was the trend function modeling non-periodic changes in the value of the time series, *s*(*t*) represented the periodic changes (e.g., weekly and annual seasonality), and *h*(*t*) represented the effects of holidays that occur at potentially irregular intervals over one or more days. Prior for holidays and events component *h*(*t*) was not reported because of being not applicable in current research. The error term *ϵ_t_* corresponded for any idiosyncratic changes that the model did not account for.

GT data stood for standardized data; it was assumed that the exhibition of saturating growth will not affect the data. For the trend model, the linear one was chosen with the following equation:(2)$$g\left(t\right)=\left(k+a{\left(t\right)}^{T}\delta_{j}\right)t+\left(m+a{\left(t\right)}^{T}\gamma_{j}\right),$$
where *k* was the growth rate, *a*(*t*) was the vector of adjustments (*ϵ* {0;1}*^S^*), the rate adjustments that occur at time *s_j_* are represented by *δ_j_*, *m* was the offset parameter. To make the function continuous, the *γ_j_* was set to −*s_j_δ_j_* (*s_j_* was the changepoint times *j* = 1, ..., *S*).

The uncertainty of the predicted trend was estimated by extending the generative model forward. The generative model for the trend implied that there were *S* changepoints over a history of T points, each of which has a rate change *δ_j_*∼*Laplace*(0, *τ*). The *τ* parameter directly controlled the model’s flexibility when modifying its rate. Simulation of future rate changes mimicking those of the past was done by replacing τ with a variance derived from the data. It was done by application a Bayesian framework with a hierarchical prior for τ to obtain its posterior, otherwise the maximum likelihood estimate of the rate scale parameter: $\lambda =\frac{1}{S}\sum_{j=1}^{S}\left|\delta_{j}\right|$ was applied. Future change points were randomly selected so that the average frequency of change points was equal to that in the past.

Modeling seasonality relied on the standard Fourier series to provide a flexible model of periodic effects [20].
(3)$$s\left(t\right)=\sum_{n=1}^{N}\left(a_{n}\cos\left(\frac{2\pi nt}{P}\right)+b_{n}\sin\left(\frac{2\pi nt}{P}\right)\right).$$

The seasonality adjustment *s* was done by constructing a matrix of seasonality vectors for each value of *t* of historical and future data.

The entire model from (1) for current project was developed in Stan code [21]. For Stan’s [22] model fitting the limited memory Broyden–Fletcher–Goldfarb–Shanno algorithm [23] was implemented to find a maximum of the a posteriori estimate.

The Prophet forecaster stage of the current project was applied with linear growth, the trend uncertainty using the MAP estimate of the extrapolated generative model (0.8), and the flexibility of the automatic changepoint selection (0.95). The change points were derived for the first 80% of the time series to provide a sufficient margin for projecting the trend forward and to avoid overfitting fluctuations before the forecasting. Data predictions were made based on the data from a period of 261 weeks (from 8 January 2017 to 2 January 2022), and the prediction period was set at 52 weeks (from 9 January 2022 to 1 January 2023). The threshold for determining the significance of the absolute value of the delta change points was set at the level of 0.1. The layers overlaying significant change points were added to the Prophet forecast. 

The cross-validation procedure was performed automatically for a set of historical cutoffs using the cross-validation function. The forecast horizon, the size of the initial training period, and the interval between cutoffs were set to 52, 150, and 26 weeks, respectively.

## 3. Results

The Ljung Box test for independence [24,25] with 26 lengths of lag (half of the forecast period) showed that time series from Google Trends were non-stationary (*p* < 0.001).

Data from the three countries surveyed showed similar dynamics of interest with slightly different levels of fluctuations and the severity of variations in trend dynamics.

### 3.1. Germany Dataset

The data fitting diagram is shown in Figure 1. Observations were marked as black dots. The blue solid line showed the predicted values, and the light blue polygons represented the uncertainty interval (80%). The vertical brown dashed lines indicated the location of the potential trend change points. The brown solid line denoted the trend.

As seen in Figure 1, more than 90% of the observation points were covered by the confidence level of 80% with cross validation prediction metrics of 15.85 for RMSE and 12.50 for MAE. From 2017 to 2022, the trend line went through three phases of change: moderate growth until mid-2018, more intensive growth until mid-2019, and then a downtrend until mid-2020. There has been a steady moderate increase in interest in Invisalign since this time. This trend is expected to continue throughout the forecast period. In 2022, according to the equation describing the fitted trend line, the predicted interest in Invisalign will increase by 0.15 points per week so, at the end of 2022, it will increase by around 13% compared to the end of 2021. The algorithm also allowed for the analysis of the demand dynamics over the course of a year. The months of February, May, and August saw the most interest in Invisalign. On the other hand, Invisalign demand was lowest in the months of June and December.

### 3.2. France Dataset

The data fitting diagram is shown in Figure 2.

As in Figure 1, more than 90% of the observation points in Figure 2 were covered by the confidence level of 80%. The model has a greater fluctuation with a cross validation prediction metrics of 22.37 for RMSE and 17.95 for MAE. The algorithm showed seven points of change on the trend line and moderate growth to 2018, slightly more intensive growth to the beginning of 2019, followed by a declining trend to mid-2020 was observed. In mid-2020, the decline was replaced with an even more intense growth that stabilized to moderate in early 2021. Since then, interest in Invisalign has increased steadily and moderately, and this linear trend is expected to continue throughout the forecast horizon. According to the equation describing the adjusted trend line, interest in Invisalign is expected to grow by 0.07 percentage points per week in 2022, so at the end of 2022, it will increase by approximately 6% over the end of 2021. In March and September, Invisalign was the most popular. In contrast, demand was lowest in April and August.

### 3.3. Italy Dataset

The data layout chart is presented in Figure 3.

For Italian data, more than 90% of the observation points in Figure 3 were covered by the confidence level of 80%. The model has a greater fluctuation with a cross validation prediction metrics of 24.94 for RMSE and 20.16 for MAE. The algorithm showed eight points of change on the trend line, similar to previous countries. There was a moderate growth to mid-2018, slightly more intensive growth to a peak in mid-2019, followed by a declining trend to mid-2020. In mid-2020, the decline was replaced by even more intense growth, which stabilized to moderate at the beginning of 2021. Since then, interest in Invisalign has grown steadily and moderately, and its linear trend is projected to continue throughout the forecast period. Based on the equation describing the adjusted trendline, interest in Invisalign is projected to increase by 0.11 percentage points per week in 2022, so at the end of 2022, it will increase by approximately 9% compared to the end of 2021. Over the months of January and September, Invisalign attracted the most attention with lower interest in August.

## 4. Discussion

The aim of this study was to predict future interest in Invisalign in the most developed countries of the European Union. This is the first study predicting interest in “Invisalign” queries in a Google search engine. An earlier study observed an increase in “Invisalign” queries globally when compared to a five-year period [6]. This trend was confirmed in the present study with a forecast for 2022 in the three most developed countries of the European Union. According to the study on global interest in Invisalign, there was a decrease in “Invisalign” queries during the first COVID-19 lockdown in spring 2020. This trend is also evident in the present study, as a decline in inquiries for “Invisalign” immediately after the COVID-19 pandemic outbreak is visible in all the European countries surveyed, with growing interest in the continuing course of the pandemic. With aligners, almost the entire planning process can be done virtually and treatment monitoring can also be done remotely [26]. Thus, in the challenging times of the pandemic, the use of aligners can ensure treatment progress. The most commonly reported urgent need is bracket breakage, which may explain growing interest in Invisalign treatment during the COVID-19 pandemic, as fewer emergency visits were needed with aligners (5.1%) compared to fixed appliances (74.7%) [27]. According to previous studies, another benefit of aligners was shorter treatment duration and chair time [28,29].

The superior comfort and esthetics of aligners may be the reason why more patients prefer this type of orthodontic treatment [29]. Study participants in England were willing to pay more for esthetic than metal braces [30]. In Italy, more than half the study population preferred aligners over fixed appliances [31]. This is consistent with the current study which predicts a 9% increase interest in “Invisalign” queries in Italy in 2022. Aligners can be removed for drinking and eating, maintaining oral hygiene is easier, and the lack of metal may reduce gum irritation and pain [5]. Aligner wear can be monitored remotely and this has been shown efficient for improving patient compliance [32]. With the projected increase in interest in aligners, more such applications and solutions may be introduced.

As interest grows in improving appearance, awareness of potential benefits, such as improved social and economic opportunities and increased self-esteem and Oral Health-Related Quality of Life (OHQOL) resulting from orthodontic treatment, are increasing. It is likely that many adults did not consider orthodontic treatment previously due to the appearance and pain of having fixed appliances. Shalish et al. examined OHQOL during treatment and documented that fixed lingual appliances were the most painful and difficult to adapt to, in contrast to aligners that were the easiest [33]. Invisalign patients were significantly more likely to feel self-confident during treatment than those with fixed appliances [34]. In a study analyzing 1564 Twitter posts about Invisalign treatment; positive tweets were about treatment satisfaction and improved self-esteem [9]. Another study showed that pain was troublesome for 16% of Invisalign patients but this did not affect the overall positive experience [35]. In the OHQOL study, patients treated with buccal brackets, regardless of whether they were metal or esthetic, reported the worst quality of life, while the group with aligners showed almost no change. However, after treatment, quality of life was found to improve in all groups [36]. The easier adaptation to removable aligners during treatment may result in a growing interest in Invisalign. Our survey predicts an increase in “Invisalign” queries in three EU countries: from 6% to 13% by the end of 2022. However, according to a systematic review with meta-analyses it appears that orthodontic treatment with aligners has worse results compared to treatment with fixed appliances [37,38]. Moreover, careful selection of patients for aligner treatment is challenging [39].

Invisalign procedures require the purchase of a scanner and certification courses. Align Technology, Inc. provides training and certification for doctors [5]. In addition, aligner companies approach universities and provide discounts on their products to encourage the postgraduate orthodontic students [30]. Universities may have to change the curriculum for students or professionals to meet future market demands. In a UK survey, 59% of undergraduate students showed interest in Invisalign courses [40]. An earlier study suggested that patients are willing to pay more for orthodontic services provided by doctors with higher levels of formal education. In a competitive market, clinicians should optimize their skills and be prepared for patient demands for different types of appliances, adjusting their skills accordingly [30]. A survey of the general population showed the perception that dental professionals provide the highest quality care, higher than that of direct-to-consumer (DTC) providers by mail. Those participants who chose DTC aligner treatment claimed they did so because of convenience, less time at the dentist, and lower price, but not for high quality of treatment [41].

The main limitation of this study is the use of only one search engine; however, Google is the most popular with a market share of 91.42% [42]. Often referred to as “google it” also may show the popularity and wide global use of this search engine. The problem of multilingual analysis does not affect our conclusions because the brand name Invisalign is the same in the investigated countries. However, the selection of the brand among aligners can be considered a limitation of the study, so the selection method was carefully performed and described in the method section of the paper. Finally, the exact design of Google algorithms is not disclosed by the company. Phrase overlap [43] or errors in the approach to questions asked by a single Google search user may also be an unavoidable aspect of such studies. Google search queries can be affected by various factors, and external influences cannot be ruled out.

## 5. Conclusions

The study predicts increased interest in “Invisalign” queries in the three most developed countries of the European Union. 

This research can help orthodontists, dentists, and national dental and orthodontic societies use information on current market trends, needs and demands, to prepare for and plan for the way dentistry will change in the coming years. 

Prediction on a population basis is difficult, but monitoring large-scale online trends can help gauge public interest in medical topics. Dentists who are informed about predictions in the oral health market can become scientifically and technologically better prepared to meet patient expectations and manage their clinics more effectively.

## Figures and Tables

**Figure 1 ijerph-19-03105-f001:**
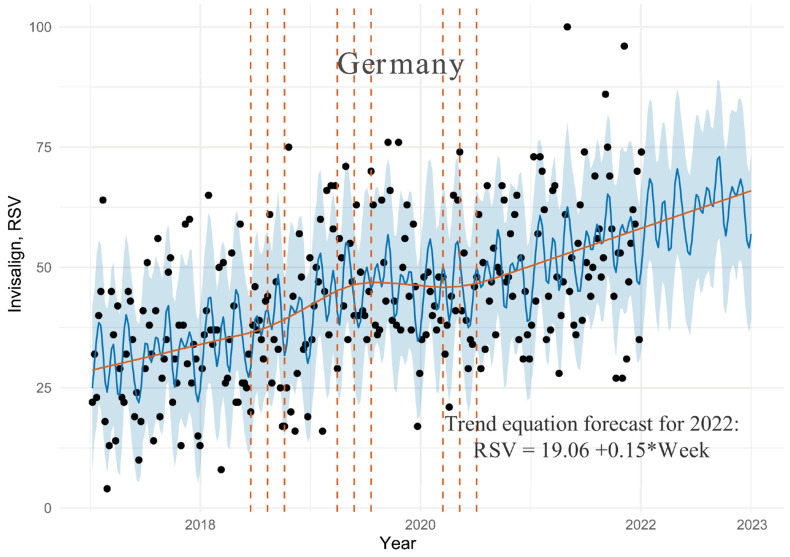
Graph of GT time series distribution for search results for keyword “Invisalign” in Germany with the fit and prediction by Prophet algorithm and trend equation forecast for 2022, where “Week” is the week number counted from the beginning of dataset (8 January 2017).

**Figure 2 ijerph-19-03105-f002:**
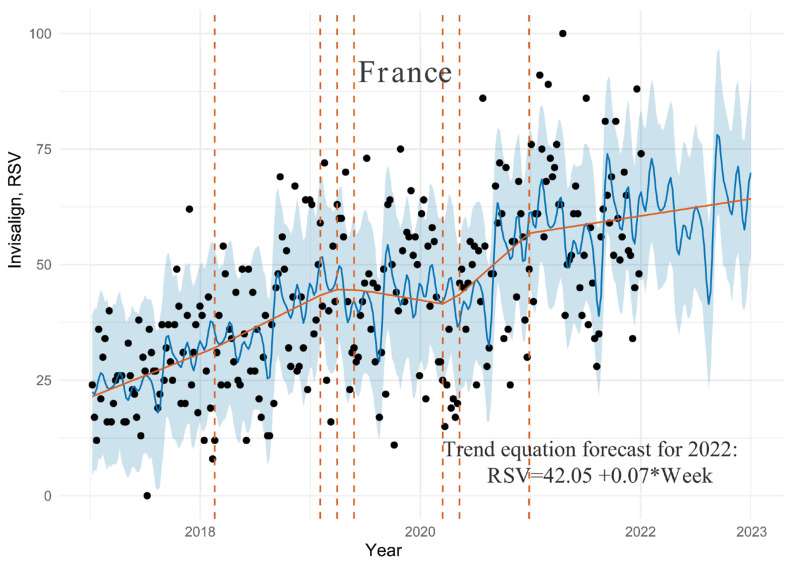
Graph of GT time series distribution for search results for keyword “Invisalign” in France with the fit and prediction by Prophet algorithm and trend equation forecast for 2022, where “Week” is the week number counted from the beginning of dataset (8 January 2017).

**Figure 3 ijerph-19-03105-f003:**
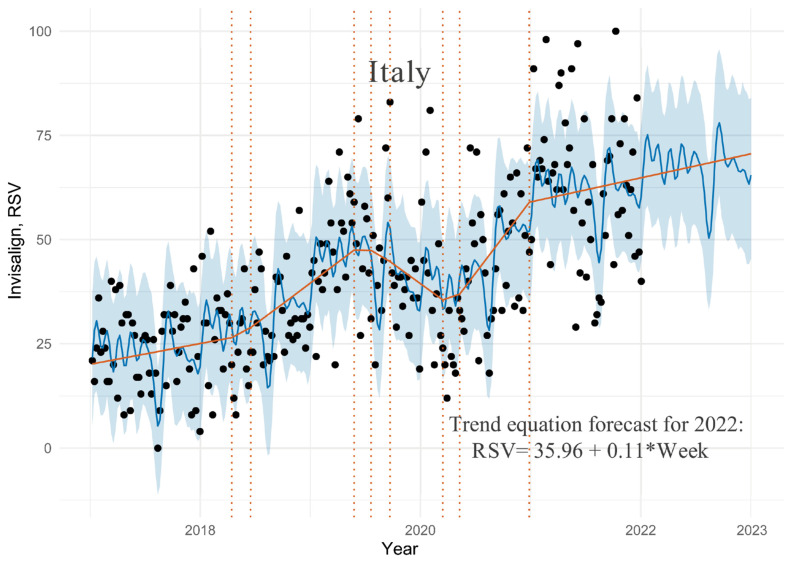
Plot of the distribution of the GT time series for the search results of the keyword “Invisalign” in Italy with the adjustment and prediction by Prophet algorithm and trend equation forecast for 2022, where “Week” is the week number counted from the beginning of dataset (8 January 2017).

## Data Availability

All data are available from corresponding author on a reasonable request.

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
