# Peer review of "Predicting Interest in Orthodontic Aligners: A Google Trends Data Analysis"

_ijerph, 2022, doi:10.3390/ijerph19053105_

Round 1

Reviewer 1 Report

The subject of the paper “Predicting interest in orthodontic aligners: a Google Trends data analysis” is timely and valuable to the audience of the IJERPH. Researchers presented results of predicting searches of „Invisalign” in three EU countries with the use of Google Trends Data.

Overall, the paper is well structured, reads quite well, and covers the existing literature quite well. The analysis of the data is interesting and well documented. However, the manuscript is not ready for publication.

I have a few major concerns about the study. Mainly with data collection and the possibility to repeat the study.

First, Google Trends does not present data about expressions. It shows data about search queries or topics. There is a significant difference between the use of search query and topic. Which one was used for this study? I noticed that Google Trends proposes two topics: „Invisalign” and „- Invisalign Aligner and Retainer”

Second, in line 84, you are writing about global country selection, yet you did not choose global countries but European countries. How did you measure the most developed economy? (Line 86)

Line 88 - 0 does not mean there were no queries. It can also mean that the number was low.

Third, what were the other setting for the Google Trends search? Which category did you choose, and what index was searched for?

The data collection description is vague and hard to repeat. This section needs to be rewritten from scratch.

The statistical part of the manuscript is sound and clear. It is easily repeatable.

In the discussion section, a few times authors compare their results with works conducted in the United Kingdom. Why was the UK not considered in this study, except not being an EU member?

I also have a concern about this study's clear limitation, which was not mentioned. The study covers five years of data. During this time number of devices connected to the internet and the number of people using the internet has increased. It is obvious that the number of searches with the use of Google Search Engine rises each year. For five years, many people started to use the internet; smartphones appeared in a huge number, very often with Android, which uses Google as the standard search engine. The increased interest is obviously correlated with an increased number of internet users and the increased number of devices connected to the network. I except that author elaborate on this as one explanation of rising trends.

Of course, I admire the amount of work put into this research. The topic is a good one, and the authors are “potentially” onto something interesting, but I do believe that the current manuscript is at an early stage to be published IJERPH. I hope you find these comments helpful as you continue to push the paper forward.

Reviewer 2 Report

Ijerph-1611972

 Predicting interest in orthodontic aligners: a Google Trends 2 data analysis. by  SyciÅ„ska-Dziarnowska  et al.

We have here a trends prediction for interests in aligners, exemplified here for Invisalign aligners, based on Google trends data forecasted using the Prophet package for rstasts.

Methodologically, the study largely seems to be in order. However, it remains unclear how the bias caused by the specific influence of the brand owners alone and/or in cooperation with Google or by local peaks in demand was considered. Since it is known that Google search queries can be influenced, external influence cannot be ruled out, so that this possibility should at least be discussed.

Moreover, the rationale for conducting such a study for a journal dealing with public health is unclear. The gain in knowledge for practitioners is negligible. The paper has more of the character of a study that could have come from the marketing department of Invisalign.

Overall, therefore, a scientific journal in the field of public health does not seem suitable to publish such data.

Round 2

Reviewer 1 Report

Thank you very much. All of my previous comments were correctly addressed. Thank you very much for claryfing how the collection data procedure was performed. I think that the manuscript has been significantly improved. I wish you good luck in your future work.

Reviewer 2 Report

Since I had already not considered the original version of this review adequate, I am now actually a little surprised to have been considered as a reviewer for the revised version.
Apparently, the authors have tried to meet the suggestions of the other reviewers. This must be appreciated, of course, but the assessment can only be made by the respective reviewers themselves.
Basically, the changes have not resulted in any improvement that would be significant enough for me to revise my original assessment.
I regret, however, that I still find the paper to be below average when measured against the competition in the field. At the same time, I would like to express that I do not wish to overemphasize my role in the first round of revision and will, of course, concur with the judgment of the other reviewers and the editor.

Author Response

Dear Reviewer,

Thank you very much for your comments.

Kind regards

The Authors